# Current State of Genomics in Nursing: A Scoping Review of Healthcare Provider Oriented (Clinical and Educational) Outcomes (2012–2022)

**DOI:** 10.3390/genes14112013

**Published:** 2023-10-27

**Authors:** Joanne Thomas, Jordan Keels, Kathleen A. Calzone, Laurie Badzek, Sarah Dewell, Christine Patch, Emma T. Tonkin, Andrew A. Dwyer

**Affiliations:** 1Genomics Policy Unit, Faculty of Life Sciences and Education, University of South Wales, Pontypridd CF37 1DL, UK; joanne.swidenbank@southwales.ac.uk; 2William F. Connell School of Nursing, Boston College, Chestnut Hill, MA 02476, USA; jordan.keels@bc.edu; 3Global Genomics Nursing Alliance (G2NA), Pontypridd CF37 1DL, UK; calzonek@mail.nih.gov (K.A.C.); lzb340@psu.edu (L.B.); sdewell@tru.ca (S.D.); christine.patch@wellcomeconnectingscience.org (C.P.); 4National Institutes of Health, National Cancer Institute, Center for Cancer Research, Genetics Branch, Bethesda, MD 20892, USA; 5Ross and Carol Nese College of Nursing, Penn State University, University Park, PA 16802, USA; 6School of Nursing, Thompson Rivers University, Kamloops, BC V2C 0C8, Canada; 7Engagement and Society, Wellcome Connecting Science, Hinxton CB10 1RQ, UK

**Keywords:** genomics, midwifery, nursing, nursing education, nursing practice, outcome measures

## Abstract

In the 20 years since the initial sequencing of the human genome, genomics has become increasingly relevant to nursing. We sought to chart the current state of genomics in nursing by conducting a systematic scoping review of the literature in four databases (2012–2022). The included articles were categorized according to the Cochrane Collaboration outcome domains/sub-domains, and thematic analysis was employed to identify key topical areas to summarize the state of the science. Of 8532 retrieved articles, we identified 232 eligible articles. The articles primarily reported descriptive studies from the United States and other high-income countries (191/232, 82%). More than half (126/232, 54.3%) aligned with the “healthcare provider oriented outcomes” outcome domain. Three times as many articles related to the “knowledge and understanding” sub-domain compared to the “consultation process” subdomain (96 vs. 30). Five key areas of focus were identified, including “nursing practice” (50/126, 40%), “genetic counseling and screening” (29/126, 23%), “specialist nursing” (21/126, 17%), “nurse preparatory education” (17/126, 13%), and “pharmacogenomics” (9/126, 7%). Only 42/126 (33%) articles reported interventional studies. To further integrate genomics into nursing, study findings indicate there is a need to move beyond descriptive work on knowledge and understanding to focus on interventional studies and implementation of genomics into nursing practice.

## 1. Introduction

Since the initial sequencing of the human genome in 2003, the “genomic era” has revolutionized our understanding of health and illness, enabled rapid diagnosis and identification of at risk individuals, and informed tailored precision therapies that have improved health outcomes. Genomic healthcare involves the use of an individual’s genomic information (i.e., genetic test results) to inform care. Importantly, genomics is a lifespan competency applicable from before birth through the end of life, including preconception/prenatal testing (for inherited conditions and chromosomal anomalies), newborn screening, disease susceptibility, screening and diagnosis, determining prognosis and guiding treatment decisions, and monitoring disease burden and recurrence [1]. As such, healthcare providers must be equipped with genomic competencies to reap the full promise of genomic discovery to improve outcomes for individuals, families, communities, and populations.

While genomic healthcare holds great promise, there is an inadequate number of trained healthcare professionals with genomic competency to meet the growing demand for genomic health care [2]. There were early calls for nursing to be involved in the burgeoning field of genomics [3]. Nurses are the most numerous of trained healthcare professionals with a global workforce of 27.9 million, including 19.3 million professional nurses [4]. Further, there is a broad range in scope of practice across nursing roles depending on academic preparation and training. For example, the advanced practice registered nurse (APRN, e.g., nurse practitioner, nurse midwife) scope of practice includes assessing, diagnosing, and treating. Accordingly, nurses with genomic competency directly increase workforce capacity for accessing and delivering genomic healthcare. Yet, to effectively deliver genomic healthcare to the public, nurses, at all levels of preparation, must have appropriate genomic knowledge and skills that underlie competency [5].

Over the past 20 years, the American Academy of Nursing and Sigma Theta Tau International have published a number of articles calling for and describing how nursing can be involved in genomic healthcare. Such system-level calls have focused on integrating genomic competencies into nursing education [6,7,8,9], in hospitals/healthcare systems [10,11,12], and in healthcare policy [7,13]. More recently, the Chief Nurse for the International Council of Nurses highlighted why genomics matters to nursing in her blog [14]. In 2012, as part of a wider project to establish a “blueprint” for genomic nursing science [5], a team conducted a systematic review to identify and assess evidence of improved patient outcomes when nursing care was delivered by nurses with genomic competencies. The specific research question was “What health outcomes are associated with nursing care which incorporates genetic and genomic principles, technology and information?” [5]. The team searched existing literature published up to May 2012, yet of the 415 retrieved articles, only 7 met inclusion criteria, precluding qualitative synthesis. Thus, nearly a decade into the “genomic era”, there was yet insufficient evidence to address the question regarding genomic nursing outcomes. The extended lag between discovery and implementation into practice, sometimes referred to as the “17-year gap”, is a widespread challenge in healthcare [15]. A number of robust, evidence-based applications support genomics in practice. Guidelines from the Clinical Pharmacogenetics Implementation Consortium (CPIC) and the National Comprehensive Cancer Network (NCCN) are relevant to nursing, and in particular, APRN practice. Thus, it seems timely to re-evaluate the current state of the implementation of genomics into nursing practice.

The aim of this study was to identify the progress of nursing and/or midwifery in genomics in the 10 years (2012–2022) since the initial mixed-methods systematic review of the literature in May 2012 (reported as Appendix A) [5]. To chart the current state of genomics in nursing/midwifery, we conducted a systematic scoping review of the literature to address the broad question “What outcomes are associated with nursing and midwifery practice that incorporates Omics research, principles, technology and information?”. Identified articles were sorted according to the Cochrane Collaboration outcome taxonomy [16]. Herein, we report findings related to healthcare provider oriented outcomes (2012–2022) and highlight future directions for nursing and midwifery in genomics.

## 2. Materials and Methods

We conducted a scoping review guided by the Arksey and O’Malley framework [17,18]. There is no registered protocol associated with this scoping review. The literature search and review was conducted using Covidence™ systematic review software (2023) [19]. The study findings are reported using the Preferred Reporting Items for Systematic Reviews and Meta-Analyses extension for the reporting of scoping reviews (PRISMA-ScR) [20].

### 2.1. Identifying the Research Question

The scoping review process was guided by a single primary question: “What outcomes are associated with nursing and midwifery practice that incorporates Omics research, principles, technology and information?”. For the purpose of this review, nursing/midwifery practice is defined as: patient/client care, patient/client counselling, clinical interventions, health promotion, research, and education that is provided or delivered by registered nurses/midwives.

### 2.2. Identifying the Relevant Literature

With the support of a research librarian, we conducted literature searches (December 2020–July 2022) in four databases (PubMed, CINAHL Plus, EMBASE, Web of Science core collection). The structured search used the medical subject headings (MeSH) terms and key words (Appendix B).

### 2.3. Selecting the Literature

Inclusion criteria for eligible studies included the following: (i) primary research studies published in a peer reviewed journal; (ii) studies reporting findings from original studies performed globally (i.e., any country of the world); (iii) studies reporting results/outcomes associated with a nursing activity in Omics (i.e., genomics, proteomics, metabolomics, metagenomics, phenomics, and transcriptomics); (iv) studies with an explicit focus on nursing/midwifery activities; (v) published in English; (vi) published since May 2012 (i.e., immediately following the publication of the original mixed-methods systematic review [5]). Exclusion criteria included: (i) review articles, letters to the editor, or commentary articles; (ii) reporting secondary or tertiary sources; (iii) studies with no clear nursing/midwifery contribution; (iv) studies with peripheral involvement of nurses/midwives (e.g., part of the study team); (v) studies in which nursing/midwifery activities are not the study focus or without defined outcomes; (vi) not published in English; (vii) published prior to May 2012. Articles retrieved from the structured literature search were imported into Covidence™ for screening. After removing duplicate titles, articles underwent independent, dual review of title and abstract (JT, JK, KAC, CP, AAD, ETT). Discrepancies were determined by a third independent reviewer from within the team. Subsequently, the remaining articles underwent independent, dual, full-text review (JS, JK). Any discrepancies during the review process were resolved by a third independent reviewer (KAC, AAD, ETT).

### 2.4. Charting the Data

Independent investigators (JT, JK) extracted data using a structured, predetermined data collection form. The structured form was developed specifically for this scoping review to capture title, authors, year, country, study population, nursing/midwife population, methods, nursing/midwife activity or intervention, genomics focus, summary of study findings/outcomes, and relevant Cochrane Collaboration outcome taxonomy (Appendix C) [16]. Briefly, the Cochrane taxonomy comprises five outcome domains (“consumer”; “health care provider”; “health service delivery”; “related to research”; and “societal or governmental”), each with respective sub-domains. Risk of bias was not conducted due to the methodological variability of the included studies.

### 2.5. Collating, Summarizing, and Reporting Results

Extracted data from included articles were organized in a master table (Appendix A). Articles were grouped according to Cochrane Collaboration outcome taxonomy domain “healthcare provider oriented outcomes” that includes two sub-domains (“knowledge and understanding” and “consultation process”). Results are reported using descriptive statistics (i.e., percentages) and narratively.

### 2.6. Synthesis of Results

To synthesize nursing/midwifery roles in Omics within the Cochrane Collaboration “healthcare provider oriented outcomes” domain, two investigators (JK, AAD) reviewed and analyzed identified articles using an iterative process to identify thematic elements [21]. Identified thematic elements were subsequently collapsed into categories across settings and target audience for more granular reporting. Subsequently, thematic analysis was applied to identify key topical areas for nursing in genomics to summarize the state of the science in the respective areas.

### 2.7. Patient and Public Involvement

There was no patient or public involvement in this scoping review.

## 3. Results

### 3.1. Selection of Sources of Evidence

The initial search strategy yielded a total of 8532 articles. Removing duplicates left 8448 articles for title and abstract screening. Screening excluded 7833 articles, leaving 615 articles for full-text review. Subsequently, 232 included articles were retained for analysis. The PRISMA flow diagram (Figure 1) depicts the review process and reasons for exclusion. A table delineating the attributes, characteristics, and key findings for each included article is provided in Appendix A.

### 3.2. Characteristics of Sources of Evidence

The 232 included studies spanned 33 countries, yet nearly half (111/232, 47.8%) of the studies were from the United States (USA) [5,9,22,23,24,25,26,27,28,29,30,31,32,33,34,35,36,37,38,39,40,41,42,43,44,45,46,47,48,49,50,51,52,53,54,55,56,57,58,59,60,61,62,63,64,65,66,67,68,69,70,71,72,73,74,75,76,77,78,79,80,81,82,83,84,85,86,87,88,89,90,91,92,93,94,95,96,97,98,99,100,101,102,103,104,105,106,107,108,109,110,111,112,113,114,115,116,117,118,119,120,121,122,123,124,125,126,127,128,129,130]. Based on the World Bank Income classification, the vast majority of studies were conducted in high income countries (191/232, 82.3%) [5,9,22,23,24,25,26,27,28,29,30,31,32,33,34,35,36,37,38,39,40,41,42,43,44,45,46,47,48,49,50,51,52,53,54,55,56,57,58,59,60,61,62,63,64,65,66,67,68,69,70,71,72,73,74,75,76,77,78,79,80,81,82,83,84,85,86,87,88,89,90,91,92,93,94,95,96,97,98,99,100,101,102,103,104,105,106,107,108,109,110,111,112,113,114,115,116,117,118,119,120,121,122,123,124,125,126,127,128,129,130,131,132,133,134,135,136,137,138,139,140,141,142,143,144,145,146,147,148,149,150,151,152,153,154,155,156,157,158,159,160,161,162,163,164,165,166,167,168,169,170,171,172,173,174,175,176,177,178,179,180,181,182,183,184,185,186,187,188,189,190,191,192,193,194,195,196,197,198,199,200,201,202,203,204,205,206,207,208,209,210,211,212]. Included studies were then classified according to the Cochrane Collaboration outcome taxonomy [213]. More than half (126/232, 54.3%) of articles related to “healthcare provider oriented outcomes”, followed by “consumer oriented outcomes” (67/232, 28.9%) and “health service delivery outcomes” (39/232, 16.8%). Herein, we report the findings relating to the predominant outcome identified in the systematic literature search (“healthcare provider oriented outcomes”).

### 3.3. Characteristics of Studies Reporting “Healthcare Provider Oriented Outcomes”

Our structured literature search (2012–2022) identified 126 articles relating to “healthcare provider oriented outcomes”. There are two sub-domains within this Cochrane outcome. Approximately three-quarters of identified articles (96/126, 76.2%) relate to the “knowledge and understanding” sub-domain [22,26,27,28,29,35,36,45,49,50,54,56,59,60,61,63,64,65,66,67,69,71,72,74,81,87,88,92,93,95,98,99,102,105,106,108,109,110,111,113,117,118,120,121,122,123,125,127,128,134,142,143,153,154,155,158,161,165,168,169,170,172,173,175,178,180,182,186,187,192,193,201,203,209,211,214,215,216,217,218,219,220,221,222,223,224,225,226,227,228,229,230], while the remaining articles (30/126, 23.8%) pertain to “consultation process” [34,38,43,73,84,94,100,131,136,138,140,144,145,151,152,167,176,179,183,189,190,194,195,197,205,206,212,231,232,233]. There was consistent, steady, and nearly linear growth of nursing genomics publications relating to “healthcare provider oriented outcomes” with an average of 11 ± 3 articles (median: 12) articles published each year from 2012 to 2022 (Figure 2).

Geographically, nearly half of studies (60/126, 47.6%) [22,26,27,28,29,31,34,35,36,38,42,43,45,49,50,54,56,59,60,61,63,64,65,66,67,69,70,71,72,73,74,79,81,84,87,88,92,93,94,95,98,99,100,102,105,106,108,109,110,111,113,117,118,120,121,122,123,125,127,128] were published by groups from the USA, followed by the Netherlands (10/126, 8%) [131,138,140,145,176,178,183,189,192,205] and the United Kingdom (UK, 9/126, 7%) [143,151,161,175,182,197,203,206,209], while the other 30 countries individually contributed to <1% of total publications.

In terms of methodology, 78/126 (62%) employed a quantitative approach [22,26,27,28,34,35,36,45,49,56,61,63,65,66,69,71,72,81,84,87,88,93,98,99,100,105,106,108,109,111,113,117,120,121,122,123,127,134,138,140,144,145,155,158,165,167,168,170,172,173,176,178,179,182,186,187,189,192,193,195,197,209,211,215,216,217,220,221,222,223,224,225,226,227,228,229,230,231]. Other methods were less frequently used, including mixed-methods (24/126 19%) [29,50,60,70,73,74,79,92,94,95,102,118,125,142,143,153,154,183,194,205,206,218,219,232], qualitative (20/126 15.9%) [31,38,42,43,54,64,67,110,128,131,136,152,169,175,180,190,201,212,214,233], descriptive (3/126, 2%) [59,161,203], and clinical audit (1/126, <1%)[151]. Identified studies were primarily non-interventional (84/126, 67%) [22,26,27,28,31,34,38,42,43,45,49,54,59,61,64,66,67,69,71,73,84,87,88,93,94,100,102,108,109,111,113,118,120,121,123,125,127,128,131,134,136,138,144,152,153,155,158,165,167,168,169,172,173,175,176,179,183,186,187,190,192,194,195,201,209,211,212,214,216,217,220,221,222,223,224,226,227,228,230,231,232,233], while 42/126 (33%) were interventional in nature, including five articles reporting on instrument development (i.e., development, testing/validation, psychometric properties) [29,35,36,56,60,63,65,70,72,74,79,81,92,95,98,99,105,106,110,117,122,140,142,143,145,154,170,178,180,189,193,197,205,206,215,218,219,225,229].

### 3.4. Settings of Articles Reporting “Healthcare Provider Oriented Outcomes”

Thematic analysis of articles on “healthcare provider oriented outcomes” identified reports spanning five settings, including “clinical practice” in work settings (85/126, 67.5%), “nursing education” in academic settings (23/126, 18.3%), “professional development” for practicing nurses (8/126, 6.3%), “academic research” (i.e., instrument development and validation) (5/126, 4.0%), and “other” (5/126, 4.0%) (Figure 3A). Articles in the “clinical practice” setting [28,31,34,38,42,45,54,60,64,69,71,72,73,79,84,87,88,93,94,95,100,102,105,108,109,111,120,121,122,125,128,131,136,138,140,143,144,145,151,152,153,154,155,158,161,165,167,168,169,172,173,175,176,178,179,180,183,187,189,190,192,194,195,197,201,205,206,209,211,212,214,216,217,218,219,220,222,223,224,226,227,229,231,232,233] primarily evaluated nurses’ knowledge, views, and attitudes, suggesting that nurses believe it is important to integrate genomics into practice. However, results suggest that nurses lack the knowledge and confidence for integration. “Nursing education” articles [22,26,27,29,49,50,56,59,65,66,67,74,98,99,106,110,118,127,134,186,193,215,225] examined genomics knowledge, comfort, and confidence among nursing faculty (n = 6) and students (including undergraduate [n = 6] and graduate [n = 3] nursing students, and one article including both faculty and undergraduate students). Key findings demonstrate limited knowledge and comfort with genomics in nursing faculty and students. In contrast, “professional development” articles [61,63,81,92,113,203,221,230] focused on educational programs for practicing nurses. Overall results suggest that nurses benefit from exposure to genomics material. “Academic research” articles [26,50,113,120,123] concentrated on the Genomic Nursing Concept Inventory (GNCI) [26,50,123] and the Genetics and Genomics Nursing Practice Survey (GGNPS) [113,120]. Psychometric properties suggest the GNCI is a reliable and valid tool to assess genomic knowledge among nurses. The GGNPS is a psychometrically evaluated instrument that evaluates nurses’ knowledge/competency as well as attitudes/receptivity, confidence, and decision/adoption of genomics in nursing practice [234]. The “other” category included heterogenous topics including development of a mobile pharmacogenomics application [70], results from a workshop on nursing and genomics [142], a Delphi study on genomics and nursing [170], and an educational framework for genomics in nursing [35].

### 3.5. Target Groups of Articles Reporting “Healthcare Provider Oriented Outcomes”

After sorting articles by setting, a subsequent round of analysis was conducted to provide more granular insight into the groups under investigation. Identified articles examined several groups/populations, including practicing nurses, nursing students (both undergraduate and graduate), and nursing faculty. Overall, the majority of studies (71/126, 56%) examined “knowledge and perceptions” of the relevance of genomics to nursing (Figure 3B). Among practicing nurses, 21 articles examined application of genomics to nursing practice (within inpatient hospitals and ambulatory clinics, palliative care, and public health settings) [49,70,71,84,102,105,131,138,145,151,161,167,183,192,195,197,209,220,229,231,233] and 10 focused on continuing education related to genomics [35,92,95,122,140,142,143,154,170,180]. Six studies centered on educating nursing faculty [36,59,65,74,81,106], and twelve centered on preparatory education of undergraduate (n = 6) [29,67,134,186,193,215] and graduate (n = 6) [98,99,110,117,118,225] nursing students. Six articles reported on “other” topics ranging from storytelling in genomics [182,203] and instrument development/validation [26,50,113] to the use of culturally appropriate pedigree nomenclature [94].

### 3.6. Current State of Genomics in Nursing across Key Areas of Focus

To summarize the current state of the science of genomics in nursing, we used thematic analysis to identify key areas of focus in the Omics nursing literature. All identified articles related to genomics. No articles were identified relating to other Omics topics. Five key areas of focus were identified, including “nursing practice” (50/126, 40%), “genetic counseling and screening” (29/126, 23%), “specialist nursing” (21/126, 17%), “nurse preparatory education” (17/126, 13%), and “pharmacogenomics” (9/126, 7%).

#### 3.6.1. Nursing Practice Outcomes

Practicing nurses were defined as licensed nurses working in a clinical setting (i.e., inpatient hospital or ambulatory practice). A total of 50 articles were classified as relating to nursing practice. The dominant Cochrane sub-domain for nursing practice articles was “knowledge and understanding” (45/50, 90%) [26,35,36,49,50,59,63,64,69,71,81,87,92,93,108,111,113,117,120,121,122,123,125,127,128,142,143,154,155,165,169,170,172,175,187,201,203,209,218,219,220,221,222,223,224]. Fewer articles (5/50, 10%) focused on the sub-domain “consultation process” [94,144,151,167,197]. Two-thirds of articles on practicing nurses were non-interventional (33/50, 66%) [49,59,64,69,71,87,92,93,94,108,111,121,125,127,128,143,144,151,155,165,167,169,172,175,187,201,203,209,220,221,222,223,224]. Five articles (10%) reported on instrument development/validation, including the Genomic Nursing Concept Inventory (GNCI) [26,50,123] and the Genetics and Genomics Nursing Practice Survey (GGNPS) [113,120]. Thematic foci of nursing practice articles included “knowledge and attitudes” (16/50, 32%) [64,87,94,108,121,125,128,155,169,172,187,201,221,222,223,224]; “nursing education” (13/50, 26%) [35,36,49,59,63,81,117,142,143,154,203,218,219]; “implementation into practice” (12/50, 24%) [71,92,111,122,144,151,165,167,175,197,209,220]; “instrument development/evaluation” (5/50, 10%) [26,50,113,120,123]; “knowledge/attitudes” and “integration into practice” (3/50, 6%) [69,93,127]; and “nursing competencies” (1/50, 2%) [170]. Results indicate misconceptions and inaccurate understanding of genomics among nurses [69,108,111,127,128,142,155,165,169,187,221,222,223,224], resulting in challenges integrating OMICs into practice [64,69,94,121,144,209]. There is a divide between nurses who see genomics as important [49,64,69,87,93,125,128,142,143,155,175] to practice and those who are uncertain of its applicability [172,201,224]. Those who view genomics as important to practice tended to be APRNs or midwives [87,125,220]. Exposure to genomics education increased knowledge and confidence among nurses [35,63,81,154,203,218].

#### 3.6.2. Genetic Counseling and Screening Outcomes

Twenty-nine articles related to genetic counseling and screening. Approximately two-thirds of articles aligned with the Cochrane sub-domain “consultation process” (19/29, 66%) [38,43,73,100,131,138,140,152,176,179,183,189,194,195,205,206,212,231,232], while the remaining articles (10/29, 34%) related to the “knowledge and understanding” sub-domain [60,61,109,153,178,180,192,214,215,216]. Studies were primarily non-interventional (21/29, 72%) [38,43,61,73,100,109,131,138,152,153,176,179,183,192,194,195,212,214,216,231,232]. Thematic foci of genetic counseling and screening articles included ”implementation into practice” (11/29, 38%) [38,109,131,138,179,180,183,189,195,231,232], “knowledge and attitudes” (10/29, 35%) [43,73,152,153,176,192,194,212,214,216], ”nursing education” (7/29, 24%) [60,61,140,178,205,206,215], and ”nursing competencies” (1/29, 3%) [100]. The vast majority (24/29, 83%) of articles reported on genetic counseling and the decision-making process for testing, either as the central focus of the article or in addition to another topic relating to genetic screening/testing (i.e., newborn screening, testing for hereditary cancer, and return of genetic test results, including incidental findings). Overall, articles found that nurses see a role for nursing in genetic counseling and screening [192,194,195,214,216,231]. However, the results suggest a lack of knowledge, communication skills, and confidence, which poses barriers to effectively reporting results to patients and supporting patients in making high-quality decisions (i.e., decisions that are informed and aligned with values and preferences) [43,100,109,153,179,180,216].

Insufficient understanding of genomics and a lack of confidence among healthcare providers can lead to situations where they are not adequately equipped to assist patients in making informed decisions, relaying test results, determining the best care management strategies, and making appropriate referrals. This, in turn, can lead to below-standard patient care. Previous genomics education or exposure to genomics in practice increased nursing knowledge and confidence in participating in the genetic counseling and screening process [60,140,178,179,206,215].

#### 3.6.3. Specialist Nursing Outcomes

Specialist nursing was defined as nurses working in specialty areas such as oncology and rare diseases (e.g., sickle cell disease, cystic fibrosis). Twenty-one articles related to specialist nursing roles. The majority of articles aligned with the Cochrane sub-domain “knowledge and understanding” (15/21, 71%) [54,56,72,88,95,105,158,161,168,211,226,227,228,229,230], while fewer articles (6/21, 29%) [34,84,136,145,190,233] related to the “consultation process” sub-domain. Articles reporting on specialist nursing were primarily non-interventional (15/21, 71%) [34,54,84,88,136,158,161,168,190,211,226,227,228,230,233]. Thematic foci of specialist nursing articles included “knowledge and attitudes” (9/21, 43%) [34,54,88,158,161,211,226,227,233], “implementation into practice” (6/21, 29%) [84,136,168,190,228,229], “nursing education” (5/21, 24%) [56,72,95,105,145], and “nursing competencies” (1/21) [230]. Specialist nurses perceive their role as essential to patient care; however, the articles suggest nurses are inadequately prepared to communicate genomic and medical aspects of diseases such as cystic fibrosis (CF), sickle cell disease (SSD), maturity onset diabetes of the young (MODY), or rare diseases [88,168,190,226,227,230,233]. Furthermore, nurses lack the confidence to provide safe and effective care [168,211].

#### 3.6.4. Preparatory Nursing Education Outcomes

Preparatory nursing education was defined as education prior to entering clinical practice—i.e., pre-licensure, pre-qualification, or pre-registration at the undergraduate (e.g., registered nurse (RN)) and graduate levels of nursing education (e.g., advanced practice registered nurse (APRN), including nurse practitioners (NP) and U.S. nurse midwives (NM)). Sixteen articles related to preparatory nursing education outcomes. The articles involved nursing faculty as well as undergraduate/graduate nursing students. All studies were classified under the Cochrane sub-domain “knowledge and understanding”. Seven of sixteen studies (43.8%) were non-interventional [22,27,66,67,134,182,186]. Articles examined nursing faculty (6/16, 37.5%) [22,27,65,66,74,106], undergraduate nursing education (6/16, 37.5%) [29,67,99,182,186,193], graduate nursing education (NP: 2/16, 12.5% [98,110]; NM: 1/16, 6.2% [225]), and one article (6.2%) examined both undergraduate students and nursing faculty [134]. Results suggest that many nursing faculty still lack confidence in implementing genomics content into nursing curricula [22,27]. Across identified articles on preparatory nursing education, nursing students and faculty report limited knowledge and comfort with genomics content [65,134,186]. However, interventional studies improved knowledge and confidence levels among nursing students and faculty [29,98,99,106,110,193].

#### 3.6.5. Pharmacogenomics Nursing Outcomes

Eight identified articles were classified as relating to pharmacogenomics in nursing. All articles related to the Cochrane sub-domain “knowledge and understanding”. Six of eight articles (75%) were non-interventional [28,42,45,102,173,217]. Thematic foci of pharmacogenomics articles included “knowledge and attitudes” (5/8, 62.5%) [42,45,102,173,217], “implementation into practice” (2/8, 25%) [28,70], and “nursing education” (1/8,12.5%) [79]. Findings indicate a poor understanding of pharmacogenomics and a lack of confidence in interpreting pharmacogenomic test results and applying findings to clinical care [28,42,45,70,79,102,173,217] It remains unclear whether or not nurses perceive pharmacogenomics as important to nursing practice. Article findings are discordant, as some indicate nurses view pharmacogenomics as important to practice [70,79] and others suggest nurses do not see pharmacogenomics as relevant to their profession [28,217].

### 3.7. Interventional Studies on Nursing in Genomics

Of the 126 identified articles (2012–2022), the overwhelming majority (84/126) were descriptive, and only 42 (33%) were interventional in nature. Thirty six (36/42, 86%) interventional studies related to the Cochrane sub-domain “knowledge and understanding” [26,29,35,36,50,56,60,63,65,70,72,74,79,81,95,98,99,105,106,110,113,117,120,122,123,142,154,170,178,180,193,215,218,219,225,229]. Of the six (6/42, 14%) articles aligned with the “consultation process”, five pertained to oncology (i.e., hereditary breast and ovarian cancer) [140,145,189,205,206] while the remaining article examined DNA collection technique [197]. The 42 interventional studies related to five key areas of genomics in nursing described above: (i) “nursing practice” (n = 17, 34% of all “nursing practice” articles) [35,36,63,81,117,122,142,154,170,197,218,219]—including five studies on instrument development/validation [26,50,113,120,123]); (ii) “preparatory nursing education” (n = 9, 56% of all “preparatory nursing education” articles) [29,65,74,98,99,106,110,193,225]; (iii) “genetic counseling and screening” (n = 8, 28% of all “genetic counseling and screening” articles) [60,140,178,180,189,205,206,215]; (iv) “specialist nursing” (n = 6, 29% of all “specialist nursing” articles) including three in oncology [95,105,145], two on sickle cell disease [56,229], and one on gene therapy [72]; and (iv) “pharmacogenomics” (n = 2, 25% of all “pharmacogenomics” articles) [70,79] (Figure 4).

Articles reporting on instrument development and validation focused on two instruments, the Genomics Nursing Concept inventory (GNCI) [26,50,123] and the Genetics and Genomics in Nursing Practice Survey (GGNPS) [113,120]. Psychometric evaluation indicates that GNCI is reliable at measuring the genomic knowledge of nurses [50]. The GGNPS has a different focus, as it was designed to evaluate competency/knowledge, attitudes/receptivity, confidence, and decision/adoption of genomics in nursing practice. It is the only validated instrument that specifically assesses these constructs in practicing nurses. The GGNPS employs a mix of multiple choice, Likert-type scales, and dichotomous (yes/no) questions, posing challenges for evaluating construct validity [113,120,234]. The instrument has undergone several refinements and meets accepted thresholds for face and content validity, test–retest reliability, and construct validity [113,120,234]. One study assessed nursing competencies [170]. The article suggested that knowledge alone is not enough to provide competent genomic nursing care and pointed to the critical importance of experiential learning [170]. Other interventions included online learning, blended learning (i.e., mix of online and in-person), and in-person learning, ranging from 2 h sessions to a year-long intervention. Overall, interventional articles demonstrate that educational programs increase nurses’ perceived knowledge of and confidence in applying genomics.

## 4. Discussion

This scoping review aimed to chart the current state of nursing and midwifery in Omics. A 2012 attempt to conduct a systematic review of nursing’s role in genomics was not possible, as only seven eligible articles were identified [5]. In the current study, we identified 232 eligible articles, of which more than half (126/232, 54.3%) aligned with the “healthcare provider oriented outcomes” domain of the Cochrane outcome taxonomy. All of the identified articles related to genomics, and no identified articles related to other Omics topics. There has been consistent, near linear growth in the number of publications on nursing in genomics (2012–2022), with the majority of studies coming from groups in the U.S. and other high-income countries. It may be that nurses in middle- and low-income countries who are involved in providing genomic testing, information, and care are not reporting their activities in the literature. Regardless, there is a need for broad international engagement of nurses in genomics to harness the full potential of genomics to improve outcomes of patients, communities, and populations globally.

Notably, a 2020 article by Tonkin and colleagues reported the pilot testing of a genomics in nursing self-assessment maturity matrix using a mixed-methods, participatory research approach with self-assessment [235]. The maturity matrix enables users to benchmark the current state of genomics integration into nursing practice for their country/organization. Further, the tool provides a framework guiding the development of strategic improvement, implementation, and evaluation of change over time. A relative strength of this strategic approach is that the participatory approach is highly flexible and can be readily adapted across settings regardless of the current state of implementation of genomics into nursing practice, thus making it highly relevant for use across international settings and divergent health systems.

Of the 126 “healthcare provider oriented outcomes” articles, three times as many articles related to the “knowledge and understanding” sub-domain compared to the “consultation process” subdomain (96 vs. 30). This finding suggests that, twenty years into the “genomic era”, much of the published nursing literature in genomics has concentrated on nurses’ knowledge, attitudes, and beliefs towards genomics. Overall, the cumulative body of work indicates that while nurses view genomics as important to practice, knowledge and implementation is lagging, reflecting the well-known 17-year lag between discovery and implementation [15]. Indeed, a 2018 study by Read and Ward found that both nursing students and faculty have limited understanding of genomics and share misconceptions about fundamental concepts [236]. This was also found in a study conducted by Coleman et al. (2014), which reported that genomics is important to integrate into practice; however, nurses felt inadequately prepared and lacked confidence in their knowledge of common genetic diseases [93]. The majority of articles (74%) identified in this scoping review focused on practicing nurses (i.e., clinical practice and professional development), while only 18% focused on nursing education (i.e., students and faculty).

It merits noting that requirements for incorporating genomics into nursing curricula vary. Within the U.K., knowledge and application of genomics is included within the Standards of Proficiency for Registered Nurses (https://www.nmc.org.uk/globalassets/sitedocuments/standards-of-proficiency/nurses/future-nurse-proficiencies.pdf, accessed on 15 September 2023) as part of Platform 2, “promoting health and preventing ill health”, and Platform 3, “assessing needs and planning care”. However, how this is translated into individual curricula of pre-registration courses is highly variable. In the U.S., the American Association of Colleges of Nursing publishes “The Essentials: Core competencies for Professionals Nursing Education”, most recently in 2021 (https://www.aacnnursing.org/Portals/0/PDFs/Publications/Essentials-2021.pdf, accessed on 15 September 2023). The “Essentials” span 10 domains for undergraduate and graduate nursing education including: (i) knowledge for nursing practice, (ii) person-centered care, (iii) population health, (iv) scholarship for the nursing discipline, (v) quality and safety, (vi) interprofessional partnerships, (vii) systems-based practice, (viii) informatics and healthcare technologies, (ix) professionalism, and (x) personal, professional, and leadership development. Notably, genomics is virtually absent from the “Essentials”, appearing only in the glossary (i.e., when defining health information technology and determinants of health) and in Domain 2.2, “communicate effectively with individuals” (2.2i “apply individualized information, such as genetic/genomic, pharmacogenetic, and environmental exposure information in the delivery of personalized health care”). Thus, there is a need for accrediting bodies to have a greater recognition of the importance of genomics to nursing.

There is a need for the discipline to implement a multi-level strategy to develop a robust and sustainable pipeline of nurses with genomic competency. Nursing must move beyond descriptive and observational studies and emphasize interventional studies that focus on integrating genomics into nursing practice. A holistic, multi-level approach should include studies that build genomic competency in nursing students who are the next generation of clinicians. In parallel, nursing must also develop faculty and practicing nurses who provide academic preparation in genomics, mentor clinical training experiences, and demonstrate integration of genomics into nursing practice.

We identified 42 interventional studies in our systematic literature search, of which 72% involved educational interventions for practicing nurses, nursing students (undergraduate and graduate), and/or nursing faculty. Such work has largely focused on knowledge and understanding of genomics. We found a paucity of articles with interventional designs evaluating how knowledge is implemented and applied to nursing practice. A critical gap identified from this scoping review is that there is little understanding of how teaching and improving genomics knowledge affects clinical practice. Of note, a 2015 study aimed to develop, implement, and evaluate a year-long genomics education intervention in 23 U.S. Magnet hospitals [92]. The Method for Introducing a New Competency: Genomics (MINC) program trained, supported, and supervised “champion dyads” (i.e., institutional administrator and genomics educator) to enhance integration of genomics into nursing practice. Assessment of satisfaction and institutional outcomes revealed variable effectiveness of champion dyads yet support the notion that such dyadic interventions focusing on education, policy, and healthcare services can increase nursing capacity in genomics.

Two articles reported on the Genomics in Nursing Practice Survey (GGNPS) instrument that evaluates nursing competency in practice [113,120,234]. One article was identified in the literature search and the second was published shortly after the search date. This is the only instrument identified in our scoping review that moves beyond knowledge and understanding to assess the application of genomics to nursing practice. Validated measures are critical for increasing the rigor of studies. Thus, using this validated instrument in future interventional studies and developing additional instruments will be important for increasing the rigor in measuring the application of genomics in nursing practice. Another important consideration for increasing the rigor relates to the reporting standards in studies evaluating genomic education initiatives [237]. Lack of harmonized reporting limits the evidence base for study replication and comparison across educational interventions. A recent study by Niselle and colleagues used a Delphi process with diverse participants to create the Reporting Item Standards for Education and its Evaluation in Genomics (RISE2 Genomics) [238]. This work helps advance the field by outlining quality reporting standards in genomics education and evaluation, thus supporting transparency and effective intervention appraisal.

Global clinical integration of genomics is lacking, largely due to a limited healthcare workforce with genomic competency. This scoping review found scant literature on genomic nursing competencies [100,170,230]. Considering the considerable work needed to build genomic nursing capacity, core competencies are a critical component of workforce development. Competencies can guide nursing education, training, and standards of care. Genomic nursing competencies have been established in several high-income countries, including the U.S. [100] the U.K. [239], Japan [142], and Europe [240]. However, repurposing competencies elsewhere requires in-country leadership and resources, and it must consider healthcare system design, infrastructure, and cultural attitudes/values. The Global Genomics Nursing Alliance (G2NA) is currently overseeing the development of global minimum nursing competencies in genomics for all nurses irrespective of education preparation, nursing role, or health service design. Such global efforts may be important for helping to accelerate the incorporation of genomics into nursing practice beyond high-income countries. This scoping review identified several future directions for the discipline to advance the integration of genomics into nursing (i.e., “healthcare provider oriented outcomes”) (Box 1). For example, a future direction may include developing studies that are: (i) grounded in nursing competencies in genomics; (ii) interventional (i.e., simulation); (iii) utilize validated instruments (i.e., GNCI, GGNPS); (iv) assess how embedding competencies affect nursing practice (i.e., longitudinal); and (v) are reported using established reporting standards (i.e., RISE2 Genomics).
Box 1Future directions to propel the integration of genomics into nursing.*Global efforts*: Expanding integration of genomics into nursing practice beyond high income countries.*Development pipeline*: Dual efforts to instill genomic competencies in practicing nurses and embed competencies into nursing education/training.*Competent workforce*: Basing workforce development on established nursing competencies in genomics.*Implementation into practice*: Shift focus from the “knowledge and understanding” sub-domain to the “consultation process” sub-domain.*Measurement*: Utilize validated instruments to measure application of knowledge and assess interventions.*Reporting*: Use reporting standards to facilitate transparency and comparability.

While beyond the scope of this paper, we recognize that there are additional considerations that merit consideration for future directions. One important aspect relating to genomics and nursing efforts relates to being responsive to stakeholders. Future work should ensure that genomic nursing practice is responsive to the needs of patients, communities, and populations nurses serve. Similarly, there are opportunities to engage with community stakeholders to co-create solutions for unmet genomic healthcare needs and bridge disparities in genomic healthcare [241]. In addition, nursing should consider emerging technologies to develop ways that nurses can use artificial intelligence, machine learning, and large language models to support genomic nursing care. We envision that integrating technology can help to develop a “high tech, high touch” approach to delivering genomic healthcare that is both effective and efficient while holding to the humanistic and person-centered ethos of nursing.

This scoping review has a number of relative strengths. We conducted a comprehensive review of the literature (2012–2022) and utilized a rigorous dual review using a well-established framework to guide the process [17,18]. In addition, to chart the data we used the Cochrane Collaboration outcome taxonomy (i.e., “healthcare provider related outcomes” domain, “knowledge and understanding” and “consultation process” sub-domains). This work has several limitations that are worthwhile to note. First, some articles may not have been included, as it was not always evident that the authors involved were nurses. Numerous articles were excluded because nurses were the study population (i.e., Nurses Health Study). Second, we did not conduct an extensive search of the grey literature. Third, we did not assess risk of bias given the methodological variability of the included studies.

## 5. Conclusions

There has been significant, steady growth in articles relating to nursing and genomics (2012–2022) compared to the first decade following the initial sequencing of the human genome. The vast majority of “healthcare provider oriented outcomes” articles are descriptive from high-income countries that report on non-interventional studies focusing on the “knowledge and understanding” sub-domain. To develop the discipline, there is a need to move beyond descriptive studies and focus on interventional studies and implementation. Such efforts will be necessary to develop a durable pipeline of nurses with genomic competencies to meet the burgeoning demand for genomic healthcare. There are opportunities to leverage international networks (G2NA) to help accelerate implementation of genomics into nursing practice.

## Figures and Tables

**Figure 1 genes-14-02013-f001:**
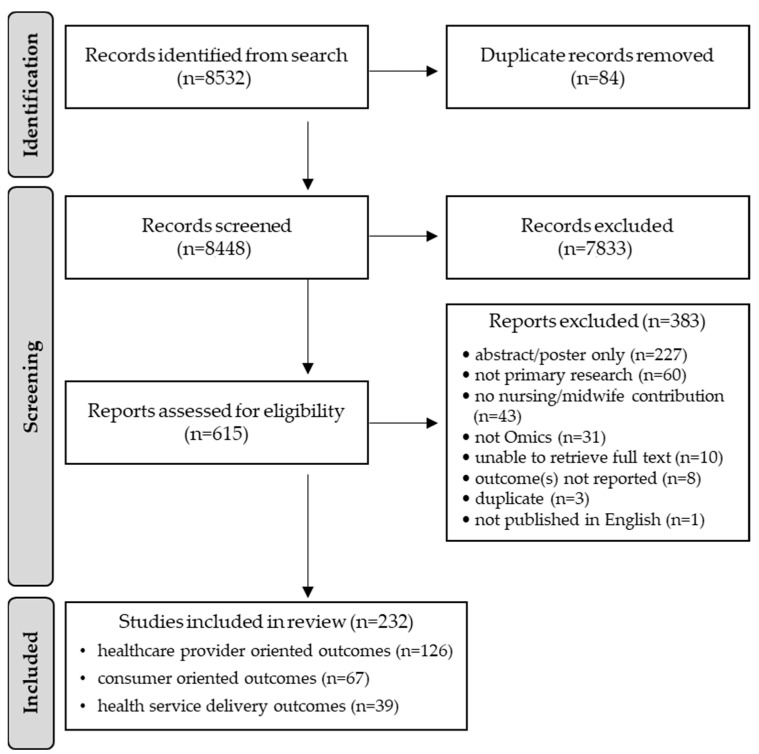
Scoping review PRISMA diagram.

**Figure 2 genes-14-02013-f002:**
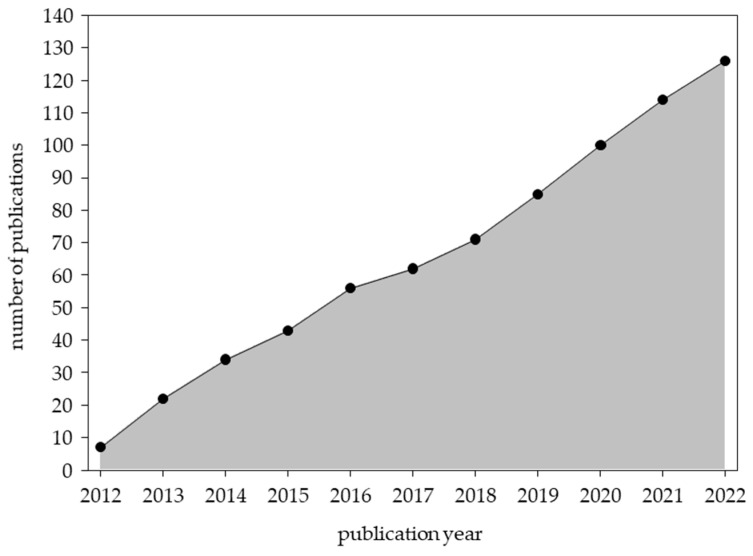
Genomic nursing publications by year (2012–2022: n = 126). A total of 126 articles were identified relating to healthcare-provider-related clinical and educational outcomes (2012–2022). On average, 11 ± 3 articles (median: 12) were published each year, exhibiting a nearly linear pattern of growth in cumulative publications on nursing and genomics.

**Figure 3 genes-14-02013-f003:**
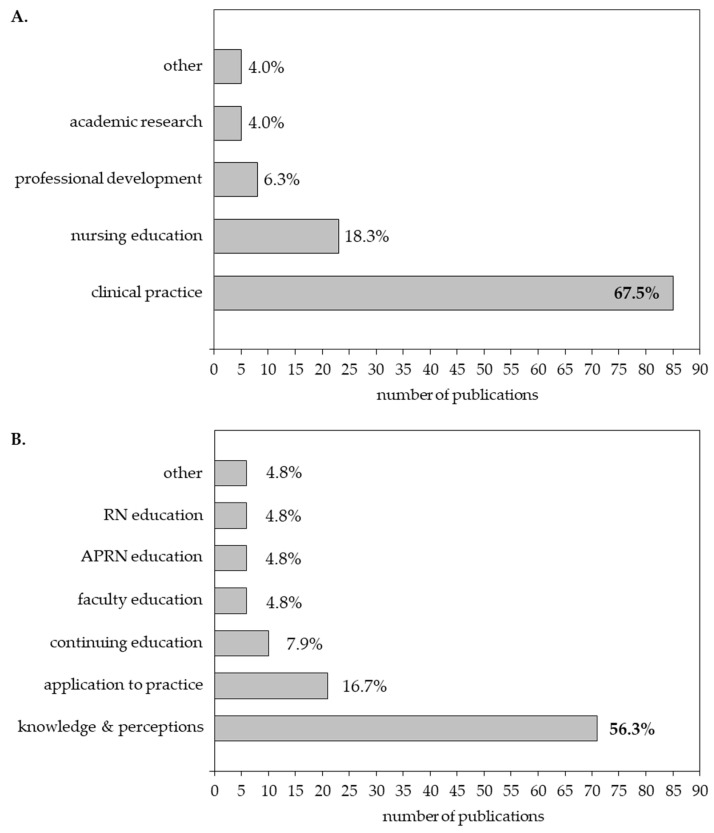
Publications by setting and topical area (2012–2022: n = 126). Top panel (**A**) depicts the number of publications by domain/setting. “Other” includes a mobile pharmacogenomics app, results from a workshop, a Delphi study, and an educational framework. Bottom panel (**B**) depicts specific topic areas of publications. Overall, 22.3% of articles focused on educating either pre-licensure registered nursing (RN) students, advanced practice registered nursing (APRN) students, nursing faculty, or providing continuing education for practicing nurses. “Other” includes articles on storytelling, instrument development/validation, and culturally appropriate pedigree nomenclature.

**Figure 4 genes-14-02013-f004:**
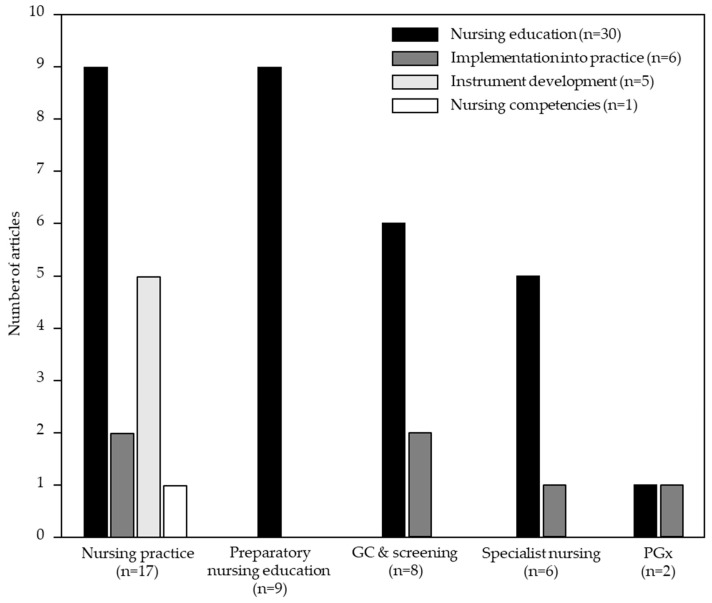
Interventional studies by area of genomics in nursing (2012–2022). Forty two interventional studies were identified across five key areas of genomics in nursing. The majority of articles (30/42, 71%) related to educational interventions across the five key areas (black bars). GC: genetic counseling; PGx: pharmacogenomics.

## Data Availability

Appendix A provides the data extraction table for all identified articles.

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
