# Peer review of "Current State of Genomics in Nursing: A Scoping Review of Healthcare Provider Oriented (Clinical and Educational) Outcomes (2012–2022)"

_genes, 2023, doi:10.3390/genes14112013_

Round 1
Reviewer 1 Report
Comments and Suggestions for Authors
Excellent scoping review! Information is comprehensive and clearly presented. Excellent synthesis of current state of genomics in nursing.
Minor suggested edits
1. Possible slight typo: author JS? Should that be corrected to JT?
2. Page 6, first paragraph: Suggest dividing into paragraphs. Lines 230-236 as first paragraph. Then start second paragraph, " In terms of methodology ..." until line 254.
Thank you-
Author Response
Excellent scoping review! Information is comprehensive and clearly presented. Excellent synthesis of current state of genomics in nursing.
- We thank the Reviewer for their positive remarks. We have responded to each comment in a point-by-point manner below.
Possible slight typo: author JS? Should that be corrected to JT?
- We appreciate the Reviewer picking this up and we have corrected this typo in the revised manuscript (highlighted text, line 159).
Page 6, first paragraph: Suggest dividing into paragraphs. Lines 230-236 as first paragraph. Then start second paragraph, " In terms of methodology ..." until line 254.
- We thank the Reviewer for this constructive comment and have created a new paragraph in the revised manuscript per the suggestion (highlighted text, line 236).
Reviewer 2 Report
Comments and Suggestions for Authors
-Are there any nursing organization or professional nursing opinion papers that can be referenced and briefly described that either call for nursing involvement in genomic care or describe how nurses should be involved in nursing care?
-given that this is, in some aspects, an update to the 2012 review, can you give a brief discussion of the findings from that review as a set up/foundation for what you find in your review?
-why did you scoping question reference "omics" and not "genomics" specifically? The title lends itself to being a genomics paper while your question and inclusion criteria included all omic areas.
-This could be asked of the editor, but the current format of your citing studies in the results narrative makes for a visually difficult read, otherwise, the authors do a commendable job but trying to succinctly describe a large number of articles that were included in your review in the results.
-Figures 1 and 2 could be added as supplementary material as you describe them narratively and to reduce the length/number of figures in the main manuscript. Similarly, it seems to me that appendix A could just be made into supplementary material as well.
-Consider updating the first sentence of the discussion as you are truly providing an updated review since 2012 and not for the last 20 years.
-given some of the deficiencies you describe in nursing education/knowledge related to genomics/pharmacogenomics, can you describe what the current requirements for genomics education is in nursing schools (at least in the United States)? Is it required component for accreditation? Are there studies you can cite that have looked at this?
-you address this somewhat in the discussion, but are there any studies where a nurse was involved in the clinical delivery of genomics care/counseling/intervention themselves and an assessment of clinical outcomes for the patient? Your discussion, as is driven by the included articles, primarily focuses on nursing knowledge, opinions and attitudes but it would be interesting to see if you found as studies that support nursing-led genomic interventions in patient care.
Author Response
We appreciate the Reviewer’s careful reading of the manuscript and constructive comments. We have responded to each in a point-by-point manner below.
Are there any nursing organization or professional nursing opinion papers that can be referenced and briefly described that either call for nursing involvement in genomic care or describe how nurses should be involved in nursing care?
- The Reviewer makes a good point. Indeed the American Academy of Nursing and Sigma theta Tau International have published several articles calling for nursing’s involvement in genomic healthcare. We have added the following to the Introduction of the revised manuscript: “Over the past 20 years, the American Academy of Nursing and Sigma Theta Tau International have published a number of articles calling for and describing how nursing can be involved in genomic healthcare. Such system-level calls have focused on integrating genomic competencies into nursing education [6-9] , in hospitals/healthcare systems [10-12], and healthcare policy [7, 13]. More recently, the Chief Nurse for the International Council of Nurses highlighted why genomics matters to nursing in her blog [14]” (highlighted text, lines 84-89)
Given that this is, in some aspects, an update to the 2012 review, can you give a brief discussion of the findings from that review as a set up/foundation for what you find in your review?
- We thank the reviewer for this comment and for the opportunity to clarify this point. As noted in the Introduction (lines 89-98): “In 2012, as part of a wider project to establish a “blueprint” for genomic nursing science [5], a team conducted a systematic review to identify and assess evidence of improved patient outcomes when nursing care was delivered by nurses with genomic competencies. The specific research question was “What health outcomes are associated with nursing care which incorporates genetic and genomic principles, technology and information?’ [5]. The team searched existing literature published up to May 2012 yet of the 415 retrieved articles only 7 met inclusion criteria – precluding qualitative synthesis. Thus, nearly a decade into the ‘genomic era’, there was yet insufficient evidence to address the question regarding genomic nursing outcomes”. The lack of published articles (n=7) was insufficient to draw any meaningful conclusions. As such, this scoping review provides a current state of the science regarding nursing and genomics.
Why did you scoping question reference "omics" and not "genomics" specifically? The title lends itself to being a genomics paper while your question and inclusion criteria included all omic areas.
- The Reviewer poses a good question. When we began this review we did not know the extent of nursing’s involvement in the field of Omics. Thus, we began the process with a broad net to capture all relevant publications. However, all the identified articles reported specifically on genomics - see line 314 in the Results (“No articles were identified relating to other Omics topics”) and lines 478-479 in the Discussion (“All of the identified articles related to genomics and no identified articles related to other Omics topics”). As all identified articles related to genomics, we used this term in the manuscript title.
This could be asked of the editor, but the current format of your citing studies in the results narrative makes for a visually difficult read, otherwise, the authors do a commendable job but trying to succinctly describe a large number of articles that were included in your review in the results.
- We agree that the citing of the articles makes for a visually cumbersome appearance. We followed the PRISMA-ScR reporting guidelines that specifies citing specific article numbers in the text. We will work with the copy editor to ensure that the final manuscript has a cleaner appearance.
Figures 1 and 2 could be added as supplementary material as you describe them narratively and to reduce the length/number of figures in the main manuscript. Similarly, it seems to me that appendix A could just be made into supplementary material as well.
- We appreciate the Reviewer’s suggestion about moving figures and Appendix A to supplemental materials. In preparing the manuscript, we closely followed the Journals’ instructions. Per the Journal instructions, Appendix A is correctly noted following the manuscript body. Similarly, the four figures are within the limit proposed by the Journal. Respectfully, we felt that it would be useful to keep them in the body of the text so the reader does not have to “toggle” back and forth between the manuscript and the online supplemental materials. If the copy editor instructs us otherwise, we will certainly follow their instruction.
Consider updating the first sentence of the discussion as you are truly providing an updated review since 2012 and not for the last 20 years.
- Per the Reviewer’s comment we have edited the sentence in the revised manuscript: “This scoping review aimed to chart the current state of nursing and midwifery in Omics” (highlighted text, lines 473-474).
Given some of the deficiencies you describe in nursing education/knowledge related to genomics/pharmacogenomics, can you describe what the current requirements for genomics education is in nursing schools (at least in the United States)? Is it required component for accreditation? Are there studies you can cite that have looked at this?
- We appreciate the Reviewer’s comment and have added to the Discussion to address this point: It merits noting that requirements for incorporating genomics into nursing curricula vary. Within the U.K., knowledge and application of genomics is included within the Standards of Proficiency for Registered Nurses (https://www.nmc.org.uk/globalassets/sitedocuments/standards-of-proficiency/nurses/future-nurse-proficiencies.pdf) as part of Platform 2 “promoting health and preventing ill health” and Platform 3 “assessing needs and planning care”. However, how this is translated into individual curricula of pre-registration courses is highly variable. In the U.S., the American Association of Colleges of Nursing publishes “The Essentials: Core competencies for Professionals Nursing Education”, most recently in 2021 (https://www.aacnnursing.org/Portals/0/PDFs/Publications/Essentials-2021.pdf). The “Essentials” span 10 domains for undergraduate and graduate nursing education including: i. knowledge for nursing practice, ii. person-centered care, iii. population health, iv. scholarship for the nursing discipline, v. quality and safety, vi. interprofessional partnerships, vii. systems-based practice, viii. informatics and healthcare technologies, ix. professionalism, and x. personal, professional, and leadership development. Notably, genomics is virtually absent from the ”Essentials” appearing only in the glossary (i.e., when defining health information technology and determinants of health) and in Domain 2.2 “communicate effectively with individuals” (2.2i “apply individualized information, such as genetic/genomic, pharmacogenetic, and environmental exposure information in the delivery of personalized health care”). Thus, there is a need for accrediting bodies to have a greater recognition of the importance for genomics to nursing” (highlighted text, lines 512-532).
You address this somewhat in the discussion, but are there any studies where a nurse was involved in the clinical delivery of genomics care/counseling/intervention themselves and an assessment of clinical outcomes for the patient? Your discussion, as is driven by the included articles, primarily focuses on nursing knowledge, opinions and attitudes but it would be interesting to see if you found as studies that support nursing-led genomic interventions in patient care.
- The Reviewer makes an excellent point. Examining nurses delivering genomics care (i.e., genetic counseling, decision support) is an important aspect of understanding the current extent to which nursing is involved in the delivery of genomic healthcare. Indeed, we fully intend to synthesize and report the articles identified in the Cochrane Domain “consumer oriented outcomes”. As noted in lines 201-206 of the revised manuscript: “Included studies were then classified according to the Cochrane Collaboration out-come taxonomy [205]. More than half (126/232, 54.3%) of articles related to “healthcare provider oriented outcomes” followed by “consumer oriented outcomes” (67/232, 28.9%), and “health service delivery outcomes” (39/232, 16.8%). Herein, we report the findings relating to the predominant outcome identified in the systematic literature search (“healthcare provider oriented outcomes”)”. Another manuscript is currently in preparation reporting a synthesis of the 67 articles reporting “consumer oriented outcomes” (i.e., effects of nurses delivering genomic healthcare) and we envision a third manuscript synthesizing the 39 articles reporting on systems aspects (i.e., “health service delivery outcomes).